# Nafion Swelling in Salt Solutions in a Finite Sized Cell: Curious Phenomena Dependent on Sample Preparation Protocol

**DOI:** 10.3390/polym14081511

**Published:** 2022-04-08

**Authors:** Barry W. Ninham, Polina N. Bolotskova, Sergey V. Gudkov, Ekaterina N. Baranova, Valeriy A. Kozlov, Alexey V. Shkirin, Minh Tuan Vu, Nikolai F. Bunkin

**Affiliations:** 1Department of Applied Mathematics, Research School of Physics, Australian National University, Canberra, ACT 2601, Australia; barry.ninham@anu.edu.au; 2Department of Fundamental Sciences, Bauman Moscow State Technical University, 2-nd Baumanskaya Str. 5, 105005 Moscow, Russia; bolotskova@inbox.ru (P.N.B.); v.kozlov@hotmail.com (V.A.K.); vutuan095@gmail.com (M.T.V.); 3Prokhorov General Physics Institute of the Russian Academy of Sciences, Vavilova Str. 38, 119991 Moscow, Russia; s_makariy@rambler.ru (S.V.G.); avshkirin@mephi.ru (A.V.S.); 4All-Russia Research Institute of Agricultural Biotechnology, Timiryazevskaya 42, 127550 Moscow, Russia; greenpro2007@rambler.ru; 5N.V. Tsitsin Main Botanical Garden of the Russian Academy of Sciences, Botanicheskaya Str. 5, 127276 Moscow, Russia; 6Laser Physics Department, National Research Nuclear University MEPhI, Kashirskoe Sh. 31, 115409 Moscow, Russia

**Keywords:** Fourier transform IR spectroscopy, swelling of polymer membrane, unwinding of polymer fibers, standardization of pharmaceutical preparation processes, Nafion, salt solution, finite-sized cell

## Abstract

When a membrane of Nafion swells in water, polymer fibers “unwind” into the adjoining liquid. They extend to a maximum of about ~300 μm. We explore features of Nafion nanostructure in several electrolyte solutions that occur when the swelling is constrained to a cell of size less than a distance of 300 μm. The constraint forces the polymer fibers to abut against the cell windows. The strongly amphiphilic character of the polymer leads to a shear stress field and the expulsion of water from the complex swollen fiber mixture. An air cavity is formed. It is known that Nafion membrane swelling is highly sensitive to small changes in ion concentration and exposure to shaking. Here we probe such changes further by studying the dynamics of the collapse of the induced cavity. Deionized water and aqueous salt solutions were investigated with Fourier IR spectrometry. The characteristic times of collapse differ for water and for the salt solutions. The dynamics of the cavity collapse differs for solutions prepared by via different dilution protocols. These results are surprising. They may have implications for the standardization of pharmaceutical preparation processes.

## 1. Introduction

The peculiarities of water induced by the strongly contrasting intrinsically hydrophobic and hydrophilic moieties of Teflon polymer continue to challenge. These peculiarities are not limited to water per se, for which some advocate the term “4th phase of water”; see the monograph [1] and references therein. Just as challenging is the nature of the nanostructures that the polymer forms. For background on Nafion™ and its properties, see the review [2]. Nafion is generated in the process of copolymerization of a perfluorinated vinyl ether comonomer with tetrafluoroethylene (Teflon), resulting in the chemical structure given below, see [2].



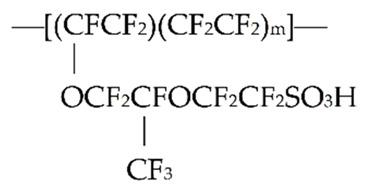



Teflon is highly hydrophobic, while the sulfonate groups are very hydrophilic. The polymer Nafion is widely studied in various fields, such as physics, chemistry, and hydrogen energetics (see, e.g., references [3,4,5,6,7,8,9,10], related to the articles on the Nafion study, issued in 2022).

While swelling in aqueous media, a nanostructure consisting of cylindrical reverse micelles forms. Water-filled channels 2–3 nm in diameter form within the Nafion membrane; see [2] for more details. These nanochannels conduct cations precisely due to the phase separation of the hydrophilic phase and the hydrophobic phase under hydration conditions. The Teflon structure provides excellent mechanical and chemical stability, while the mobile protons produced by the dissociation of the sulfonic acid groups provide active sites for cation conduction [11,12,13,14]. With the further addition of water, global packing constraints due to the size of the channels formed in the bulk medium then force an outgrowth of fibers from the membrane surface. The interior aqueous cores of the channels are negatively charged due to the dissociation of terminal sulfonate groups via R−SO_3_H + H_2_O ⇔ R−SO_3_^−^ + H_3_O^+^, with the protons residing inside the aqueous channels. This nanostructure is the key to spatial separation of H^+^ and OH^−^ ions in low-temperature hydrogen power plants [15]. The mechanism of such separation was comprehensively studied; see, for example, the review [16]. It is clear, however, that an H^+^ ion may pass through the membrane and combine with an OH^−^ ion on the other side to form water, so it is not correct to speak of a complete spatial separation of cations and anions by the membrane. The formation of the structure, consisting of the nanometer-sized channels, requires a winding up of strands into helical rods reminiscent of those that form from chiral molecules. The nanochannels can then rearrange into secondary structures inside the membrane.

When a Nafion plate is soaked in an aqueous suspension containing colloidal microparticles, an exclusion zone (EZ) appears [1]. Colloidal particles are excluded from the region adjacent to the surface of the Nafion membrane. The width of the exclusion zone *X*_0_ is of the order of 300 μm. This is astonishing and not explicable by standard theories of physical chemistry. Some physical mechanisms that might be responsible for the EZ formation are discussed in a recent review [17]. As was shown in our work [18], when Nafion swells in water, polymer fibers unwind into the liquid bulk, but they remain attached to the polymer surface and form a structure like a dilute loose brush. Furthermore, as can be inferred from our experiments on the birefringence of the exclusion zone [19], the unwound polymer fibers form a structure like that of a colloidal crystal, composed of like-charged cylindrical rods (the formation of colloidal crystals was described in the review [20]). Therefore, we have to understand the nature of the unwinding of the polymer fibers in the liquid bulk to make sense of the EZ formation.

The anomalous phenomenon of the unwinding of the polymer fibers, more reminiscent of a growth of strings of giant seaweed kelp from the ocean bed, is the source of the curious exclusion zone associated with the polymer. Subsequent experiments [21] have explored the way of growing the polymer fibers. To do that, Fourier transform IR spectroscopy was used on a Nafion-membrane, on both sides of which was a fixed volume of water in a cell. The width of the cell was much less than the extension of unwound fibers that would take place if the polymer was allowed to swell indefinitely. The unwinding fibers then abut against the cell windows. Thus, additional viscoelastic stresses arise, which forces water out of the extended polymer region between the membrane surface and the cell window. The water is expelled, and an air cavity is formed. In this case, micro-rheological effects should appear, as anticipated, for example, in monographs [22,23]. A recent work [24] attempts to probe such micro-rheological properties of the Nafion membrane by occluding macroscopic biomolecules. The effects of polymer membrane swelling on such biomolecules may not show up in normal, i.e., unconstrained conditions. We are interested in features of the transition of the extended polymer fibers from a (presumed) hydrophobic to a hydrophilic state induced by the constraints.

At the same time, the extended nanochannels emerging from the bulk Nafion have a hydrophobic surface initially. Therefore, they have to rearrange their structure, as they swell to accommodate this energetically unfavorable situation. It is clear that the fibers that fill the exclusion zone cannot be the same swollen nanochannels that exist in the bulk membrane and have a hydrophobic exterior. We could anticipate that the interaction of solute molecules with constrained polymer fibers will induce peculiarities in the local nanostructure of such aqueous solutions that might reveal insights into this. One of these will be the nucleation of nanobubbles of dissolved gas that contain highly active free radicals, as with enzymatic action [25]. Of further relevance to our study, we note that external influences on water and aqueous solutions of a pulsed electric field can lead to experimentally observed changes in the properties of water, see, for example, [26]. In other recent works, we studied the interaction of the unwound polymer fibers with suspensions of various amino-acids in physiological solutions [27]. In addition, the combined effect of static and alternating magnetic fields also leads to experimentally observed changes in the bulk properties of water, see [28,29]. The ability of water to change its functional characteristics under the influence of weak and ultra-weak magnetic field is also known [30]. Plasma discharge treatment is also an effective way to change the properties of water [31]. Mechanical effects on water deserve special attention. It is a complex and little understood physical process that changes the physicochemical properties of water; see, e.g., [32,33,34]. These effects have been known for a very long time and ignored because they were incomprehensible. An outstanding example of this goes as far back as 1948 to the work of [35]. The awareness of such intriguing effects has been revived of late, see, e.g., [36,37,38,39,40].

Our point in dwelling on these matters is important and hardly recognized. That is, as we shall show, the method of preparation of a water-solute mixture can apparently lead to pronounced changes in the properties of the mixture as compared to another preparation, but with only a single mechanical action applied to it: for example, one that consists in the cyclic mixing of 1 part of water subjected to mechanical stress, with 99 parts of unperturbed water, accompanied again by mechanical action. Explicitly, the apparently specific properties of water that has passed multi-cycle sample preparation depends on the number of cycles performed [39]. Some of these changes must be due to the initiation of reactive nanobubbles induced by the physical perturbation, see, e.g., [41,42]. But why this is so, and why and how “water structure” is affected by external physical influences, is still an open question. Nafion, with its highly contrasting “hydrophobic-hydrophilic” character providing perturbation/stresses in water gives us a probe that may provide some insights into these questions.

To summarize, external physical influence greatly affects the properties of water and aqueous solutions. Intensive mixing can cause changes in the physicochemical parameters of water. In the past, such a claim would have been thrown out of court without a hearing, on par with homeopathy. But classical theories of physical chemistry ignore the effects of dissolved gas and nanobubble aggregates, which are an emerging game changer. With these considerations on Nafion in mind, it seemed reasonable to study salt solutions, successively diluted by water that, in turn, has been subject to external physical influences. It is of interest also to investigate how the properties of the treated liquid samples depend on the type of water used for dilution (multi-cycle mechanically treated, or untreated water).

In studying these salt induced effects, we keep in mind the still unexplained and extraordinary phenomenon of bubble-bubble fusion inhibition. For some ion pairs, gas bubbles will not fuse above a salt concentration of 0.17 M, precisely the concentration of salt in the blood [43]. For other ion pairs no such inhibition occurs [44,45,46,47].

Therefore, for this work, we have investigated the dependence of the properties of a salt solution on the type of water used for its dilution, as reflected by the different behaviors they induce in Nafion. (The two types of water used multi-cycle preparation, with or without mechanical action). They also differ in the method (protocol) of preparing salt solutions of a required concentration. In carrying out the experiments we assume that solutions with the same salt concentration, but prepared according to different protocols, will interact differently with the unwound strained Nafion polymer fibers. Our research confirms the validity of this assumption.

## 2. Materials and Methods

For our experiments, we used Nafion N117 plates (Sigma Aldrich, St. Louis, MI, USA) with a thickness of *L*_0_ = 175 μm and an area of 1 × 1 cm^2^. The experiments were carried out on an analytical Fourier spectrometer FSM 2201 (LLC Infraspec, St. Petersburg, Russia). The setup is illustrated in Figure 1. The spectrometer had the following characteristics: the total spectral range is 370–7800 cm^−1^ (1.3–27 μm), spectral resolution is 1.0 cm^−1^, absolute error is ±0.05 cm^−1^. The moment when the liquid was poured into the cell set the reference time. Each measurement included 15 consecutive records of IR spectra with subsequent averaging and took 40 s (taking into account the subtraction of the background absorption due to air humidity). The time interval between each measurements was 5 min. During these intervals, the cell was removed from the spectrometer and cooled down until reaching room temperature, since the samples were slightly heated during FTIR measurement due to the absorption of IR radiation. Thus, all spectra recordings were performed under room temperature.

In the experiments the Nafion plate was placed in a cell with CaF_2_ windows; this material is transparent to infrared radiation over a spectral range of λ = 0.13–10 μm. In Figure 2 we exhibit the cell with a liquid sample (element 12 in Figure 1). The distance between the cell windows was *L* = 200 μm; a Nafion plate with a thickness of *L*_0_ = 175 μm was placed inside the cell. The red circles in the photo Figure 2 mark the inlet and outlet holes, through which the liquid was poured/flew out. The Nafion plates had a round shape, and the distance between inlet/outlet of the cell exceeds the diameter of these plates. Thus, the Nafion plate does not impede the flow of liquid from filling the cell, and liquid flows fell on the Nafion plate from above and below, and also filled the volume of the cell, free of the Nafion plate. Immediately after filling the cell, a cavity free of water is formed. The Nafion fibers unwound into the liquid are schematically shown in the figure. Note that the volume of the liquid poured into the cell was always greater than the volume of the cell itself. Despite the presence of a cavity formed during the filling process, the excess water is expelled from the outlet. After filling, the inlet and outlet holes were closed with Teflon plugs, but not tightly, i.e., there was a possibility of air to access the cell. We specifically verified that liquid does not evaporate during the experiment.

In the experiments, the value of the transmittance *K* = *I/I*_0_ was measured, where *I* and *I*_0_ are the intensities of the transmitted and incident light, which, according to the Lambert—Bouguer—Beer law [48], are related as
(1)$$I\left(t\right)=I_{0}\exp\left(-\kappa \int_{0}^{L}C_{w}(t,x)dx\right)\approx I_{0}\exp\left(-\kappa \langle C_{w}\left(t\right)\rangle L\right),$$
where *κ* is the extinction coefficient, and *L* is the distance between the cell windows. Hence, it is possible to estimate the concentration of water 〈*C_w_*(*t*)〉 via the formula
(2)$$\langle C_{w}(t)\rangle =\frac{\left|\ln K_{\min}(t)\right|}{\kappa L},$$
where *t* is the soaking time of the Nafion plate, and angle brackets mean averaging over the length *L* of the liquid layer in the cell along the IR beam (remind that *L* is the distance between the cell windows).To estimate the extinction coefficient *κ* for the liquids under study, it is necessary to carry out experiments to measure the transmittance for different *L* in the absence of Nafion, that is, when the transmitted beam intensity *I* satisfies the formula $I=I_{0}\exp\left(-\kappa L\right).$ In Figure 3, we show a typical example of a transmittance spectrum *K* for doubly distilled water, poured into a cell with the length *L* = 180 μm. For the spectral range 1.8 < λ < 2.2 μm the spectral minimum *K*_min_ is related to λ = 1.93 μm. For short swelling times *t* and small distances *L* we have *K*(λ = 1.8 μm) ≈ 0.7. Upon increasing *t* and *L* the value of *K*(λ = 1.8 μm) slightly decreases, while *K*(λ = 2.2 μm) decreases more strongly. The decrease in *K* at the wavelengths λ = 1.8 and 2.2 μm is basically due to the contribution from a more intense absorption band of water, centered at λ = 3 μm, see, e.g., [49]. Since we are interested in the quantity |ln *K*_min_| at the wavelength λ = 1.93 μm, it makes sense to count the value of *K*_min_ from that level, which is the same to all spectrograms. Indeed, since |ln *K*_min_| at *K*_min_ < 1 is a very steep function, the inaccuracies in finding *K*_min_ should result in large errors of the value |ln *K*_min_|. This is why we put hereinafter *K*(λ= 1.8 μm) = 0.7 for all spectra.

In Figure 4 we present the results of measuring |ln *K*_min_| for water vs. the distance *L* = 180, 190, 200, 210, and 220 μm. The experimental dependences are the result of averaging over five consecutive measurements. The choice of the minimum value of *L* = 180 μm is explained by the thickness of the Nafion plate *L*_0_ = 175 μm; the choice of the maximum value *L* = 220 μm is due to the fact that in this case the intensity *I* of the transmitted beam approaches zero, i.e., the results of measurements become incorrect. The value of the coefficient of variation C_v_ (the ratio of the confidence interval to the mean value) was calculated for each experimental point. The dependence |ln *K*_min_| vs. *L* is fairly well approximated by a linear function Y = 0.027 + 0.019·X, i.e., we obtain for the extinction coefficient that *κ* ≈ 0.019 μm^−1^. The value of |ln *K*_min_| for dry Nafion was also measured. The absorption in this case is due to water molecules encapsulated inside the nanometer-sized cavities in the polymer matrix (see Ref. [2]). Thus, we obtain that for dry Nafion |ln *K*_min_| = *κ*·(*C_w_*)_0_*L*_0_, where (*C_w_*)_0_ is the concentration of water in dry Nafion. In accordance with our estimations, the concentration of water in dry Nafion (*C_w_*)_0_ = 0.174.

We first studied the swelling modes of a Nafion plate in the cell with the distance between the windows *L* = 200 μm for deionized water with an initial resistivity of 18 MΩ×cm at 25 °C, prepared with different Milli-Q (Merck KGaA, Darmstadt, Germany) devices. The purpose of these experiments was to obtain the reference dependencies, which should be further compared with the dependences for aqueous solutions at different ionic concentrations.

In Figure 5 we show a typical example of a transmittance spectra *K* for Nafion, swelling for 70 < *t* < 100 min in water, with an interval of 5 min. It is seen that *K* decreases smoothly upon soaking. The decrease in the transmittance *K* is due to the collapse of the cavity shown in Figure 2, and is associated with an increase in the amount of water on the path of the IR beam as the cavity collapses. We performed visual observations of the cavity behavior in parallel with the measurements of the transmittance *K*. These observations show that the cavity, formed immediately after filling the cell with liquid, collapses from the peripheral regions to the center of the cell.

In addition, aqueous solutions of NaCl, KCl, NaClO_4_ and NaClO_3_ (Sigma-Aldrich, St. Louis, MI, USA) with various ionic content were studied. The idea behind these experiments is as follows. The motivation is that Na^+^ and K^+^ behave so differently inside and outside of biological cells that they have both strongly hydrophobic and hydrophilic moieties, and these ions affect bubble-bubble interactions specifically and dramatically above and below the physiological concentration (~0.17 M), see [44,45,46,47].

Finally, aqueous solutions of NaCl (Sigma-Aldrich, St. Louis, MI, USA) with a final concentration of 1 M, prepared according to two different protocols 1 and 2 (see Table 1), were studied. When preparing NaCl solution according to protocol 1, one volume part of deionized water sample was mixed to one volume part of 2 M NaCl solution. The technology of preparation of the water sample consisted in cyclic mixing of 1 part of water, subjected to mechanical action, with 99 parts of intact (non-processed) water; this process was also accompanied by mechanical action. The samples of deionized water with a resistivity of 18 MΩ×cm at 25 °C were obtained on a Milli-Q device. Mechanical action was performed according to the technique described in [39], and consisted in vortexing a liquid sample for 1 min at a frequency of 30 Hz and amplitude ~1 mm using a Heidolph Multi Reax (Schwabach, Germany) vortex mixer 545-10000-00; this processing was performed at room temperature without direct sunlight exposure. Subsequently, the liquid sample was shaken by oscillations in a vertical plane with an amplitude of 10 mm and frequency of 5 Hz for 30 s using an IKA orbital shaker (IKA^®^-Werke GmbH & Co. KG, Staufen, Germany), in which the platform was oriented vertically. We used water that was subjected to a different number of technological cycles: 6, 12 or 30 (hereinafter referred to as C6, C12 or C30, respectively).

According to protocol 2, one volume part of water sample (intact water, or water sample after multicyclic processing) was mixed to 99 volume parts of 1.01 M NaCl solution. For the first cycle of technological water treatment, 9.99 mL of solvent (deionized water) and 0.11 mL of the dissolved substance (also deionized water) were added, after which the sample was subjected to mechanical action. All subsequent cycles were carried out as follows: 9.99 mL of solvent (deionized water) and 0.01 mL of deionized water, obtained in the previous cycle, were added to the same sample, and then the sample was subjected to mechanical action. Intact water and water, obtained in the course of multi-cycle preparation (C6, C12 or C30), were prepared on the same day. NaCl solutions were prepared using the processed water, and were produced on the same day.

As already mentioned, we investigated the dependence 〈*C_w_*(*t*)〉 of the concentration of the liquid under study, where *t* is the soaking time of the Nafion plate; the quantity 〈*C_w_*(*t*)〉 was calculated based on Equation (2). Time start corresponds to the moment of pouring the liquid into the cell, shown in Figure 2, i.e., we can assume that at *t* = 0 the Nafion plate is practically dry. As is seen in this Figure, free of liquid cavity is formed in the cell; in the photo the cavity is lighter. The darker area corresponds to the water layer, which covers the border of Nafion plate. If there were no collapse of the cavity, i.e., if the Nafion plate was initially completely covered with water, the average water concentration along the beam path in the cell would remain constant in time. This is true for soaking Nafion in deuterium-depleted water (DDW; see [21] for more details). For DDW the effect of unwinding polymer fibers is absent, and therefore a cavity in the cell of limited size is not formed. As is seen in the photo Figure 2, the peripheral regions of the Nafion plate are always covered with the liquid. Therefore, initially hydrophobic polymer fibers, unwound in a liquid bulk, start swelling from the border of the cavity towards its center. The fibers, covered with water, acquire hydrophilic properties, which ultimately leads to the collapse of the cavity. Remember that the transmittance *K* = *I/I*_0_, controlled by the IR absorptivity of water along the beam path, was the basic parameter, explored in our experiments. When the cavity collapses, 〈*C_w_*(*t*)〉 increases along the IR beam path, and the value of *I*(*t*) decreases, see Equation (1). For each liquid sample, six consecutive measurements of 〈*C_w_*(*t*)〉 were made. The dependences obtained for the solutions were compared with the reference curves obtained for Milli-Q water; all graphs were received at the same day.

Statistical processing was carried out by using the RStudio program (Version 1.3.1093 © 2009–2021 RStudio, PBC) with R package version 4.0.3. The homogeneity of the dispersion was assessed by the Bartlett test. If the dispersion was homogeneous, Tukey’s correction analysis was used to compare multiple groups. If the dispersion was inhomogeneous, the Kruskal-Wallis analysis was used to compare multiple groups. Pairwise comparisons were made using a Student’s *t*-test (with Welch’s approximation). Differences between groups were considered to be statistically significant at *p* < 0.05, where *p*-value is the probability of obtaining the same value of statistics (arithmetic mean, median, etc.) for a given probabilistic model, provided that the null hypothesis is true., see, e.g., [50].

## 3. Results

In Figure 6 we exhibit the dependences of the average water content 〈*C_w_*(*t*)〉 for samples of water obtained on different days on different Milli-Q devices.

As follows from the graphs in Figure 6, within the first 100 min of soaking, it makes no sense to calculate the experimental errors: during this time, the points on the graphs practically coincide, and only at the times where *t* > 100 min does the spread between the extreme points exceed 1% in the concentration of water. We can claim that for deionized water within the first 100 min of soaking, the 〈*C_w_*(*t*)〉 dependence remains linear. Since the data for all tested samples were compared with the data for deionized water, the measurement time (soaking time) for these samples was limited to *t* = 100 min.

In Figure 7 we exhibit the results of measurements of 〈*C_w_*(*t*)〉 for aqueous solutions of NaCl, KCl, NaClO_4_ and NaClO_3_ for concentrations of 10^−3^, 0.1, 0.17 and 2 M for soaking times 0 < *t* < 100 min; we provide the graphs of the dependence 〈*C_w_*(*t*)〉 for the same solutions and for concentrations of 10^−2^, 0.2 and 1 M in Appendix A; these are Appendix A respectively. Experimental points correspond to averaging over six successive measurements; confidence intervals are indicated. In addition, data are given for reference water; in this case, confidence intervals are not shown, since for water with high accuracy the experimental points lie on a straight line, see Figure 6.

As follows from the graphs, the addition of ions leads to a deviation from the straight line dependence for 〈*C_w_*(*t*)〉. At the same time, for the concentration of 10^−3^ M, the differences between salts under study lie within the confidence intervals, that is, for these concentrations, the specific effects of Na^+^, K^+^, Cl^−^, ClO_4_^−^ and ClO_3_^−^ are practically not manifested. Note that for all salt solutions at 10^−3^ M the collapse rate of the cavity is higher than in water. Note also that at concentrations above 0.1 M the differences in the rate of collapse of the cavity (actually, in the behavior of 〈*C_w_*(*t*)〉) begin to reveal. At the concentrations of 2 M, the collapse rate for the NaClO_3_ monotonically increases with the swelling time, while for the KCl, NaCl, and NaClO_4_ salts, the collapse rate is a non-monotonous function of time. Now it is necessary to find a numerical criterion to characterize the difference between the studied salt solutions and water. In our opinion, the value *S*_exp_ − *S*_RW_, where *S*_exp_ and *S*_RW_ are accordingly the areas under the graph for the test mixture and the reference water, can be chosen as such a criterion. Areas *S*_exp_ and *S*_RW_ are defined as the sum of corresponding ordinate values; thus we obtain *S*_RW_ = 4.898 a.u.

In Figure 8 we demonstrate the dependences of *S*_exp_ − *S*_RW_ vs. the ionic concentrations for aqueous solutions of NaCl, KCl, NaClO_4_ and NaClO_3_. In this case, the standard deviations (SD) are of the order of the size of the point on the graph, and therefore the SD values are not shown in this graph.

In Figure 9 we show the dependences of the 〈*C_w_*(*t*)〉 value averaged over six repetitions for NaCl (1 M) solutions prepared by dilution with water (NaCl + water), according to both protocols (see Table 1). In Figure 10 we exhibit the dependences of the 〈*C_w_*(*t*)〉, also averaged over six repetitions of NaCl (1 M) solutions prepared using centesimal dilutions of water according to both protocols; the number of such dilutions was 30 (in Table 1 this is related to the designation NaCl + water C30, i.e., the initial concentration was reduced to 100^−30^ times). We also present the 〈*C_w_*(*t*)〉 data for deionized water; in this case, the scatter of the experimental points can be neglected, and the error bars are not indicated. Similar dependences of 〈*C_w_*(*t*)〉 were obtained for all NaCl solutions listed in Table 1.

As shown above, the increase in 〈*C_w_*(*t*)〉 is due to the collapse of the cavity shown in Figure 2. Recall that this cavity is formed by unwound polymer fibers which abut against the cell windows, and water is pushed out of the area between the Nafion plate and the cell window due to rheological effects. In the case of a rectilinear dependence 〈*C_w_*(*t*)〉 (deionized water), we can assume that the collapse rate is constant. As follows from the graphs in Figure 7, Figure 8, Figure 9 and Figure 10, the addition of the ionic component violates the rectilinear behavior of 〈*C_w_*(*t*)〉, i.e., in this case, the collapse rate of the cavity begins to depend on the soaking time.

For a quantitative comparison of the graphs in Figure 9 and Figure 10 with the graph for water in Figure 6 we will use the criterion of difference *S*_exp_ − *S*_RW_. As mentioned earlier, *S*_exp_ is the area under the curve 〈*C_w_*(*t*)〉 for the salt solution, and *S*_RW_ is the area under the curve 〈*C_w_*(*t*)〉 for the reference water curve. Remember that the areas *S*_exp_ and *S*_RW_ are defined as the sum Σ of corresponding ordinate values, i.e., for water *S*_RW_ = 4.898a.u.Using statistical methods (see Section 2), the differences between the studied mixtures were analyzed in terms of the 〈*S*_exp_–*S*_RW_〉 value; angle brackets mean averaging over *n* = 6 repetitions of measurements for each test solution. The measurement results are presented in Table 2 as the mean 〈*S*_exp_–*S*_RW_〉 ± standard deviation (SD).

As is seen in Table 2, although the NaCl concentration is the same in all mixtures, the behavior of the Nafion in mixtures prepared with water as a solvent that has undergone various mechanical actions is different. This fact indicates that the interactions of such solutions with polymer fibers unwound in the liquid bulk have a certain specificity, which depends on the protocol of the solution preparation. Thus, differences were found between the NaCl + water (C12) mixture and the NaCl + water mixture (solutions were prepared according to protocol 2, that is, the final mixtures were prepared by diluting the initial NaCl solution by a factor of 1.01). In addition, the dynamics of cavity collapse revealed differences between NaCl + water (C30) solutions prepared according to protocols 1 and 2. Thus, polymer fibers unwound in liquid bulk interact differently with an ionic solution, having the same final concentration. The differences, apparently, are due to the technology involved in the preparation of the water as well as different ratios of the volume fractions of the initial ionic solution and water.

## 4. Discussion

It is obvious that the peculiarities of the interaction of the ionic solutions prepared according to different protocols can manifest themselves only upon local contact with hydrophobic polymer fibers; in the bulk of the liquid far from the surface of the polymer membrane, such effects would be impossible. As shown in our work [18], the size of the area filled with unwound polymer fibers in the cell of an unlimited volume is about 300 μm. These fibers are initially hydrophobic. This is why, during the unwinding, that the surface of the polymer membrane becomes rougher and is being covered by gas nanobubbles, which, in turn, prevents the membrane from coming into contact with the liquid. Indeed, the collapse rate can depend on the degree of roughness due to the fact that nanometer-sized gas bubbles appear on the tips and rough segments of a solid substrate (see Ref. [51] for more details). Nanobubbles adhere to fibers and penetrate into the liquid bulk in the process of unwinding. If the system “Polymer membrane–water” is limited in size so that the area of unwound fibers is less than 300 µm, it will induce the shear stresses, which gives rise to the rheological effect; see [23,24,25]. Attraction forces should arise between the (conducting) polymer fibers unwound into the bulk of the water due to the appearance of additional free energy of the interaction between fibers of
(3)$$E\left(a,R\right)\approx -\frac{\hbar \omega_{p}a}{8\sqrt{2}\pi R{\left[\ln\left(R/a\right)\right]}^{3/2}},$$

Here, *R* is the distance between unwound polymeric chains, *a* is the radius of the polymeric chains, and *ω_p_* is plasma frequency. This formula is valid for the case *a* >> *λ_D_*, where *λ_D_* = (*εkT*/(8π*e*^2^*n*))^1/2^ is Debye screening length, *ε* is dielectric permittivity, *T* is temperature, and *n* is volume number density of ions. In the case of 1 M salt solution, *λ_D_* is about 3 Å, i.e., the condition *a* >> *λ_D_* is met, for more details see the derivation of Equation (40) in Ref. [52]. This interaction arises due to correlations in charge fluctuations and leads to the appearance of non-additive very long-range forces. The attractive cooperative forces are opposed by the repulsive electrostatic double layer forces between the fibers. As suggested in [19], the combination of repulsive electrostatic forces and attractive forces due to the interaction (2) could lead to formation of the spatial structure of Nafion rod-like (cylinder) strands close to the polymer surface.

We do not know the value of *R* and *a*; however, as an estimate, we can use the data of [53], where, in particular, the Nafion profile at the boundary with water was studied with atomic force microscopy, and it was found that polymer fibers in water near the membrane surface are, in fact, oriented predominantly perpendicular to the surface, and the characteristic size of the near-surface inhomogeneity is about 1.85 nm, see Figure 6b in the work [53]. This apparently should be close to the diameter of the cylinders *a*. Assuming *R* ≥ *a*, the force of attraction between polymer fibers will be very large due to the denominator in Equation (3). Further analysis requires taking into account the change in the distance *R* between the polymer fibers, which should manifest itself during the unwinding of the fibers and the formation of a field of shear stresses and deformations during elastic contact of the fibers with the windows of the cell. Since the peripheral regions of the membrane are always inside the liquid (see photo in Figure 2), the membrane swells from the peripheral areas to the center, i.e., the cavity eventually collapses. However, the dynamics of the collapse of the cavity is sensitive not only to the concentration of ions in the solution, but also to the protocol of preparation of the solution. Further research in this area will make it possible to determine the key properties of solutions that affect the dynamics of the cavity collapse. The differences between the values of 〈*S*_exp_–*S*_RW_〉 allow us to assert that the ionic solution near the polymer membrane with unwinding polymer fibers has a number of features that cannot appear in the bulk liquid. The same conclusions might be drawn on the behavior of the glycocalyx of cell membranes and the endothelial surface layer. The polymers of the glycocalyx are sulphonated, and the blood consists in mostly NaCl salt at 0.17 M.

It is natural to assume that the specific effect of various ions in our case will manifest itself precisely due to the inhibition of the coalescence of gas bubbles at concentrations above 0.17 M for NaCl and KCl salts and the absence of the inhibition effect in the entire concentration range for NaClO_3_ and NaClO_4_ [44,45,46,47]. However, as follows from the graphs in Figure 7 and in Appendix A, there are no significant differences in the behavior of the *S*_exp_–*S*_RW_ value for these salts in the concentration range of 10^−3^–2 M, and the dependences of *S*_exp_–*S*_RW_ on the ion concentration for KCl and NaClO_4_ salts practically coincide. Apparently, it should be recognized that the phenomena we are studying have no direct connection to bubble-bubble fusion inhibition.

## 5. Conclusions

If we take the view that solutions are homogeneous liquids and their properties are determined only by the concentration of a substance in them, then we expect that the properties of solutions with the same concentration of a substance, prepared using a solvent that is the same from a chemical point of view, should be identical. However, as our results show, if the liquid molecules are located between the strained polymer fibers unwound in the bulk of the liquid, then the properties of the solutions (in the context of interaction with polymer fibers) may differ even if these solutions have the same concentration.

We have shown that the rate of transition of strained polymer fibers from the hydrophobic to the hydrophilic state (that is, in fact, the rate of collapse of the cavity shown in Figure 2) depends on the protocol of preparing the final solution, i.e., on the method of obtaining the final concentration of the salt and the number of cycles of mechanical action on the water used. Thus, strained polymer fibers unwound in the liquid bulk interact differently with aqueous saline solutions having the same concentration, but prepared according to different protocols. This result should then have implications for the standardization of pharmaceutical preparation processes. Note that our results develop the already known concepts (see, for example, [54,55,56,57]) that the physicochemical properties of water (pH, electrical conductivity, surface tension, size of optical inhomogeneities, concentration of molecular oxygen, etc.) can change after the application of multi-cycle mechanical action on water.

Finally, we have shown that it is possible to change the physicochemical properties of salt solutions with the use of technologically processed solvent. This implies the theoretical possibility of controlling chemical reactions that occur in the body when using drugs. Thus, we can talk about new directions in pharmaceuticals.

## Figures and Tables

**Figure 1 polymers-14-01511-f001:**
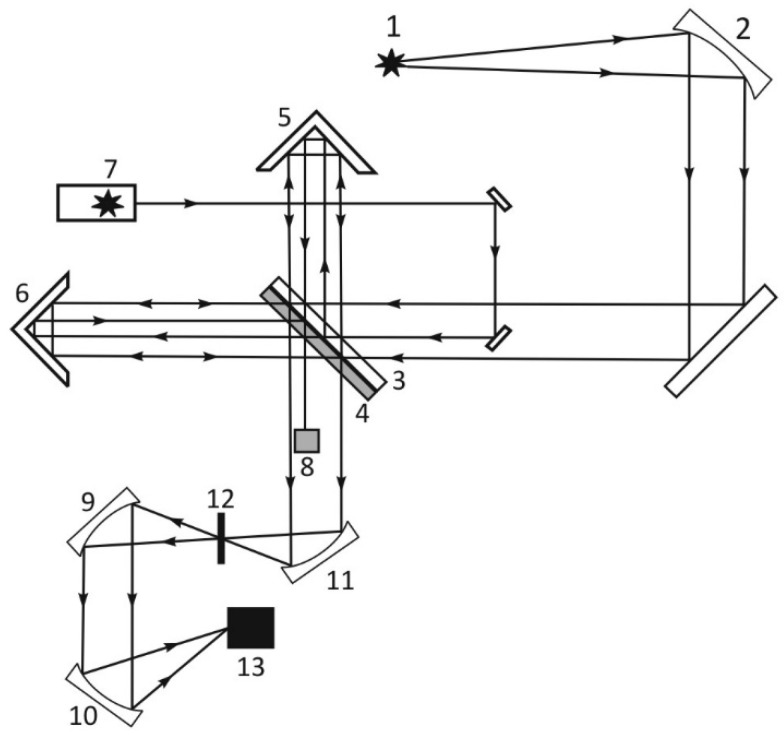
Schematic of Fourier spectrometer FSM 2201. 1—Source of IR radiation; 2, 9, 10, 11—Off-axis parabolic mirrors; 3, 4—Beam splitter and compensator (transparent in the IR range); 5—Fixed reflector; 6—Movable reflector; 7—He-Ne laser; 8—Receiver of laser radiation; 12–Cell with liquid sample; 13—IR receiver.

**Figure 2 polymers-14-01511-f002:**
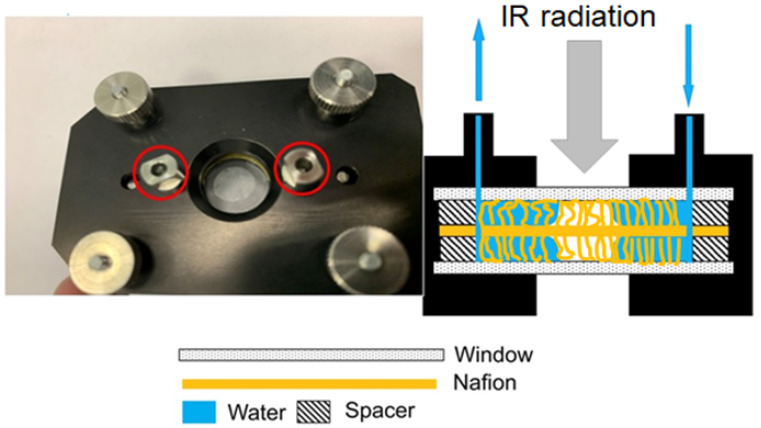
The cell immediately after filling with deionized water; the inlet and outlet are marked with red circles and blue arrows.

**Figure 3 polymers-14-01511-f003:**
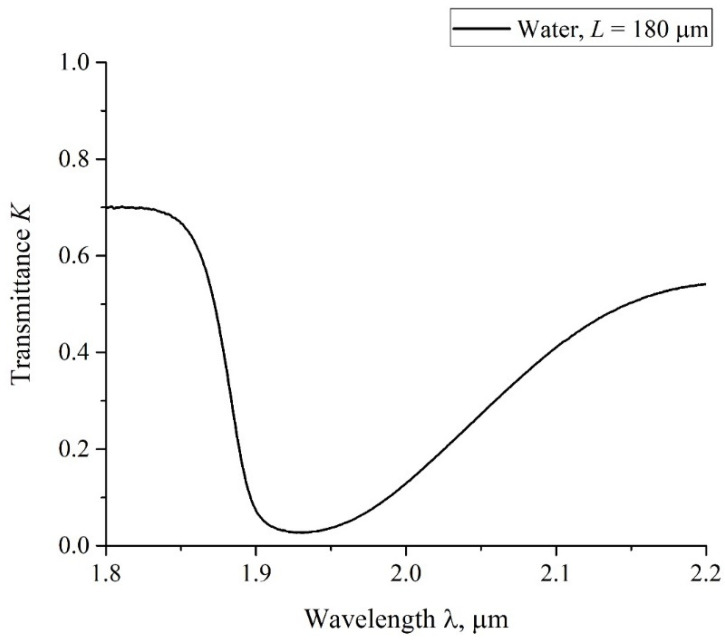
The spectrum *K* for water in the spectral range 1.8 < λ < 2.2 μm; *L* = 180 μm.

**Figure 4 polymers-14-01511-f004:**
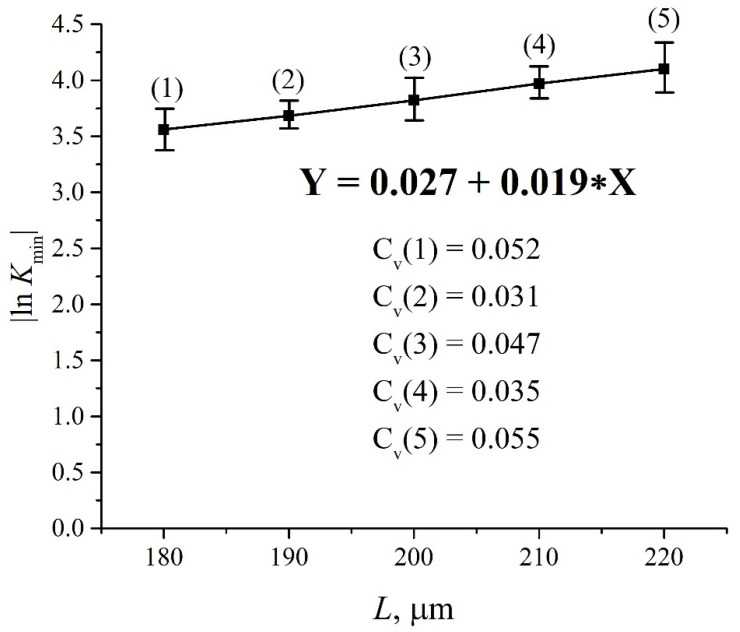
Dependences |ln *K*_min_| vs. *L* for deionized water.

**Figure 5 polymers-14-01511-f005:**
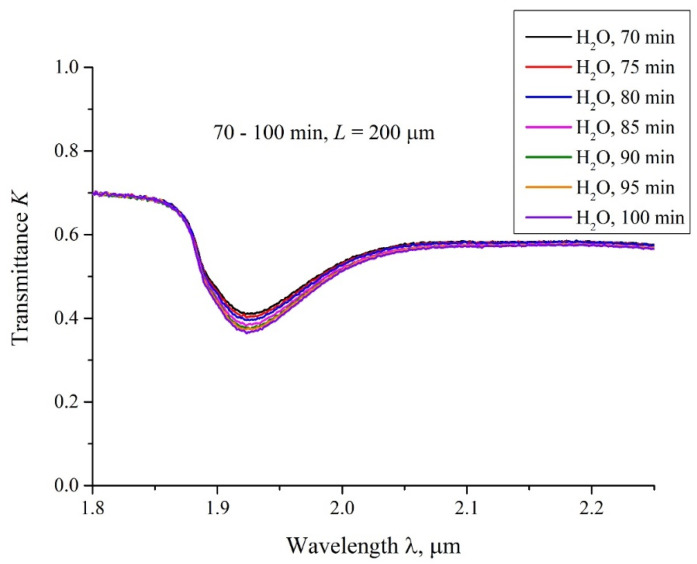
The transmittance spectra in the range 1.8 < λ < 2.2 μm for the case of swelling Nafion in ordinary water; the distance between the windows *L* = 200 μm, the curves are related to the swelling times *t* = 70, 75, 80, 85, 90, 95 and 100 min.

**Figure 6 polymers-14-01511-f006:**
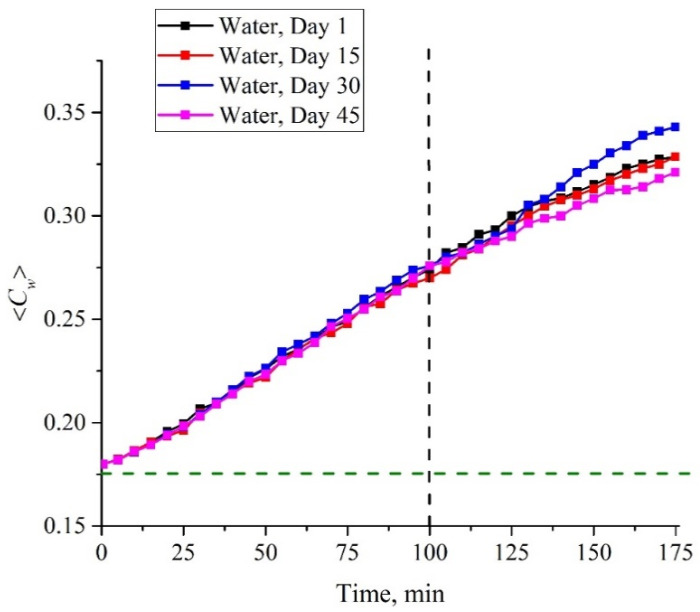
Dependences of the average water content 〈*C_w_*(*t*)〉; measurements were taken on different days at intervals of 15 days and samples were taken on different Milli-Q devices.

**Figure 7 polymers-14-01511-f007:**
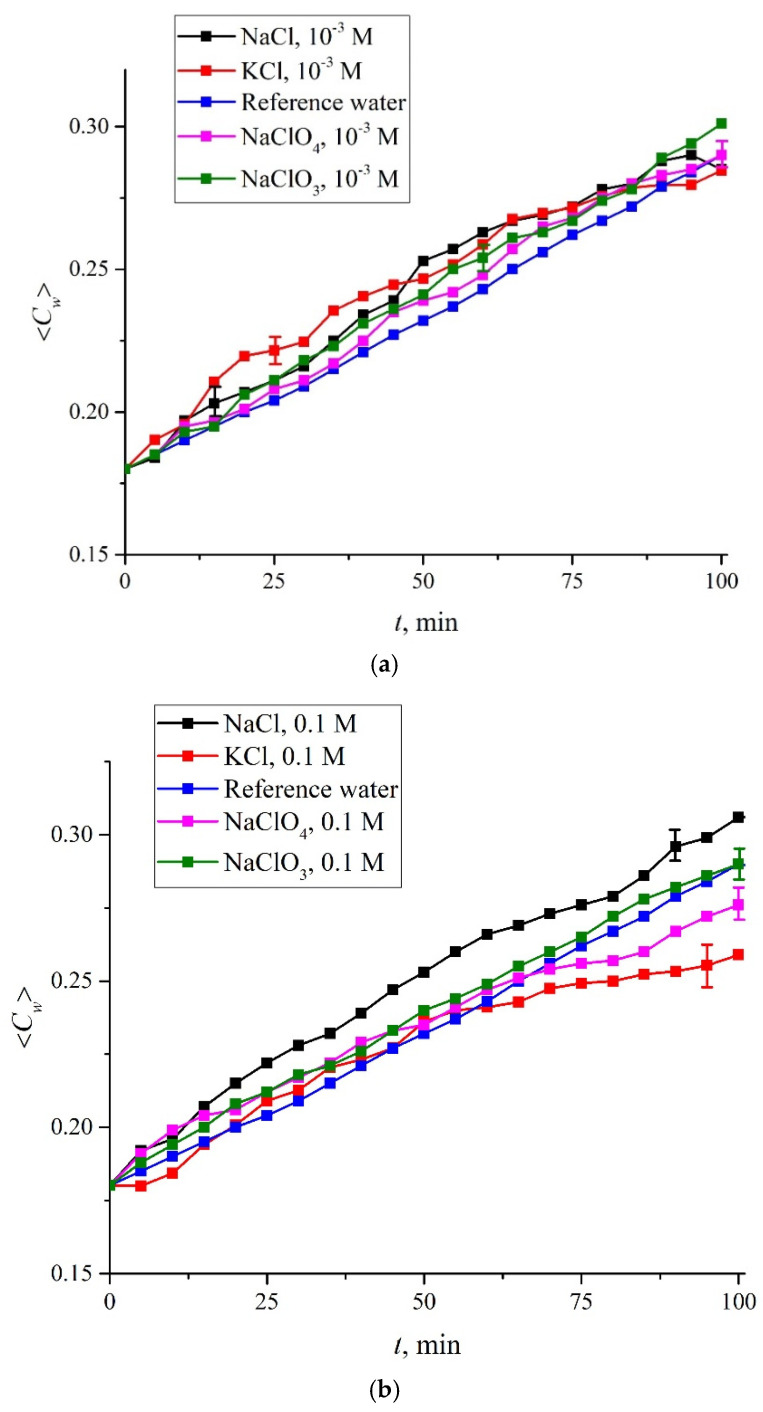
Dependence of the average concentration 〈*C_w_*(*t*)〉 for NaCl, KCl, NaClO_4_ and NaClO_3_ solutions; the dependence 〈*C_w_*(*t*)〉 for deionized (reference) water is also shown. The salt concentrations are the following: panel (**a**)—10^−3^ M; panel (**b**)—0.1 M; panel (**c**)—1.07 M; panel (**d**)—2 M.

**Figure 8 polymers-14-01511-f008:**
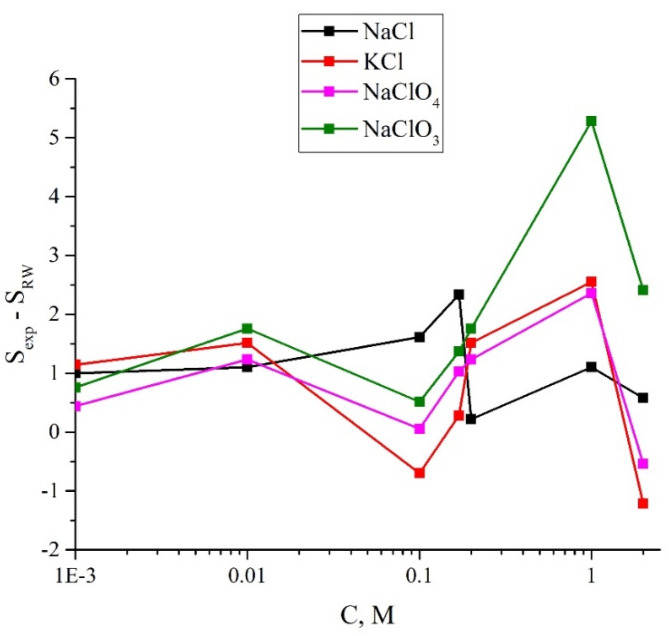
The *S*_exp_ − *S*_RW_ dependences vs. the ionic concentrations for aqueous solutions of NaCl, KCl, NaClO_4_ and NaClO_3_.

**Figure 9 polymers-14-01511-f009:**
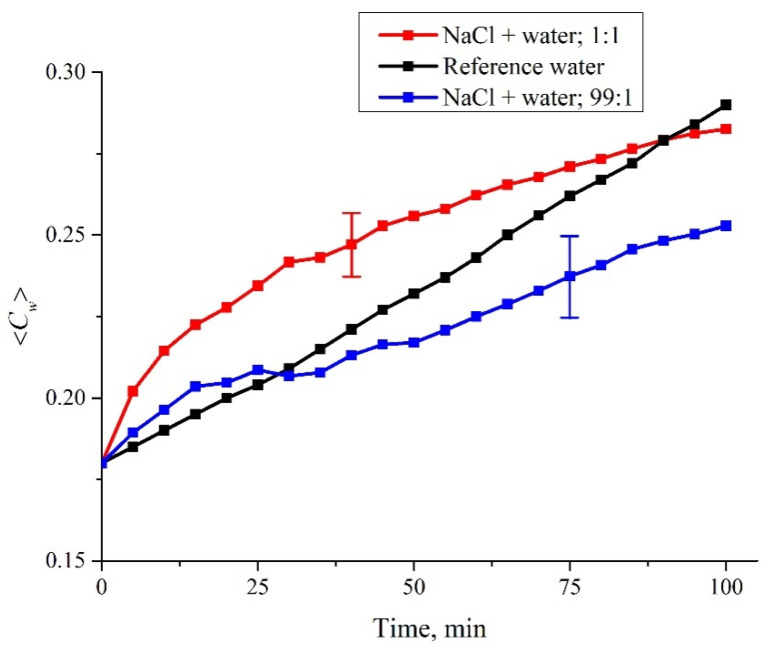
Dependence of the average concentration 〈*C_w_*(*t*)〉 for NaCl (1 M) solutions prepared by dilution with water according to the first (1:1) and second (99:1) protocols. The dependence 〈*C_w_*(*t*)〉 for deionized (reference) water is also shown.

**Figure 10 polymers-14-01511-f010:**
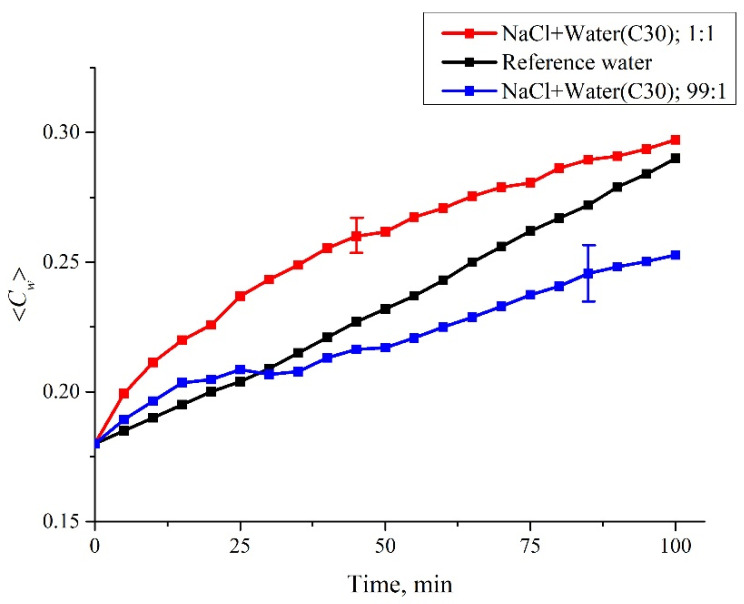
Dependence of the average concentration 〈*C_w_*(*t*)〉 for NaCl (1 M) solutions prepared by C30 dilution of water, according to the first (1:1) and second (99:1) protocols. The dependence 〈*C_w_*(*t*)〉 for deionized water is also shown.

**Table 1 polymers-14-01511-t001:** Test solutions.

Protocol 1 for Preparation of NaCl Solution (Final Concentration 1 M).		Protocol 2 for Preparation of NaCl Solution (Final Concentration 1 M)	
Sample number	Mixture components (*v*/*v* 1:1)	Sample number	Mixture components (*v*/*v* 99:1)
1	NaCl (2 M) + water	1	NaCl (1.01 M) + water
2	NaCl (2 M) + water (C6)	2	NaCl (1.01 M) + water (C6)
3	NaCl (2 M) + water (C12)	3	NaCl (1.01 M) + water (C12)
4	NaCl (2 M) + water (C30)	4	NaCl (1.01 M) + water (C30)

**Table 2 polymers-14-01511-t002:** Values of the 〈*S*_exp_–*S_RW_*〉 in the test solutions.

Test Solution (Mixture)	Volume Ratio of Mixing Components 1:1	Volume Ratio of Mixing Components 99:1
	Average 〈S_exp_ − S_RW_〉 ± SD	Average 〈S_exp_–S_RW_〉 ± SD
NaCl + water	1.71 ± 0.52	−1.36 ± 1.39
NaCl + water(C6)	2.52 ± 0.82	1.08 ± 0.34
NaCl + water(C12)	1.58 ± 0.87	2.93 ± 0.91 *
NaCl + water(C30)	2.38 ± 0.18 **	−0.25 ± 0.77 **

*—Statistically significant difference from the “NaCl + water” group (analysis of variance with Tukey’s correction, *p* < 0.05). **—Statistically significant difference from a similar solution prepared by mixing the components in a different volume ratio (Student’s *t*-test (Welch’s approximation), *p* < 0.05).

## Data Availability

The data presented in this study are available on request from the corresponding author.

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
