# Peer review of "Nafion Swelling in Salt Solutions in a Finite Sized Cell: Curious Phenomena Dependent on Sample Preparation Protocol"

_polymers, 2022, doi:10.3390/polym14081511_

Round 1
Reviewer 1 Report
In this manuscript, the features of Nafion nanostructure in several electrolyte solutions that occur when the swelling is constrained to a cell of size less than a distance of 300 μm are investigated. The constraint forces the polymer fibres to abut against the cell windows. The dynamics of the collapse of the induced cavity is studied in detail. The polymer behavior in various solutions was investigated with Fourier IR spectrometry. It is found that the dynamics of the cavity collapse differs for solutions prepared via different dilution protocols. These results may have implications for the standardization of pharmaceutical preparation processes.
I consider the content of this manuscript will definitely meet the reading interests of the readers of the Polymers journal. Therefore, I suggest giving a minor revision and the authors need to clarify some issues or supply some more data to enrich the content.
- Abstract and Introduction
- For the Keywords, ‘Nafion’, ‘salt solution’, and ‘finite-sized cell’ should also be added to attract a broader readership and highlight the significance of this work.
- Please pay attention to grammar problems, especially the missing or redundant definite articles. I suggest double-checking. I will point out several examples, but unfortunately, I cannot point out all of them. For example, in Line 79, ‘Some remarks on parallels with a physiological analogue.... are discussed in the Supplementary Materials section...’; Line 109, ‘Mechanical effects on the water deserve special attention... It is a complex and little understood physical process that changes the physicochemical properties of water...’; Line 122, ‘Some of these changes must be due to the initiation of re-active nanobubbles induced by the physical perturbation...’, and so on.
- Line 41, ‘Teflon is very hydrophobic, while the sulfonate groups are very hydrophilic. On swelling in aqueous media, a nanostructure consisting of cylindrical reverse micelles forms. Water-filled channels 2 - 3 nm diameter form within the Nafion membrane’.
There is nothing wrong with the description here. However, as a cation exchange membrane material, ‘Nafion provides microchannels for cation conduction due to the phase separation of hydrophilic phase and hydrophobic phase under hydration conditions. The Teflon structure provides excellent mechanical and chemical stability, while the mobile protons produced by the dissociation of sulfonic acid groups provide active sites for cation conduction [Solid State Ionics 319 (2018): 110-116; Electrochimica Acta 378 (2021): 138133]’. These important details and cation conduction mechanisms should be briefly mentioned here.
- Line 49, ‘This nanostructure is the key to spatial separation of Н+ and ОН– ions in low-temperature hydrogen power plants [3, 4].’
Is this due to the Donnan exclusion/repulsion effect? And in H2/O2 fuel cells, Nafion separates H+ and OH- from each half-cell as a separator. But finally, H+ may pass through the membrane and combine with OH- on the other side to form water. So the description should be modified to be more accurate. I do not consider H+ and OH- ions are separated forever and without any combination. Hence, the current description may lead to certain misunderstandings.
- Line 96, ‘ Those have a hydrophobic exterior.’This short sentence should be merged with the former or the latter sentence.
- Line 101, ‘... we note that external influences on water and aqueous solutions of a pulsed electric field can lead to experimentally observed changes in the properties of water, see, e.g., [14].’
In the introduction, ‘see [reference number]’ appears multiple times. I suggest whether it is possible to briefly describe the general description found in other literature, otherwise the Introduction part provides too little effective information, and readers have to look through lots of literature to understand the research progress of related topics.
- Results
- Page 10 and Page 14, both the figures are called Figure 6, which is definitely the wrong name.
- Line 318, ‘In Figure 7 we exhibit shows the results of measurements of the áCw(t)ñ value for...’Firstly, ‘exhibit shows’ only need one verb here. Secondly, Figure 7 demonstrates nothing about Cw(t), but Figure 6 on Page 14 shows the relevant results. I suggest the author check throughout the manuscript, whether the description of each figure in the main text is really matching the caption of the corresponding figure.
- Line 356, ‘In order to obtain a numerical criterion for the difference between the studied salts and water.’It seems this sentence is not completed yet, only half is described. In order to reach some targets, what is going to be done?
Author Response
We are grateful to the referee for a careful reading of the manuscript and very valuable comments. The manuscript has been completely rewritten in accordance with the comments and recommendations of the reviewer. Below we present our responses to the reviewer's comments, which are in italics.
For the Keywords, ‘Nafion’, ‘salt solution’, and ‘finite-sized cell’ should also be added to attract a broader readership and highlight the significance of this work.
Thanks for this remark. This flaw was corrected.
Please pay attention to grammar problems, especially the missing or redundant definite articles. I suggest double-checking. I will point out several examples, but unfortunately, I cannot point out all of them. For example, in Line 79, ‘Some remarks on parallels with a physiological analogue.... are discussed in the Supplementary Materials section...’; Line 109, ‘Mechanical effects on the water deserve special attention... It is a complex and little understood physical process that changes the physicochemical properties of water...’; Line 122, ‘Some of these changes must be due to the initiation of reactive nanobubbles induced by the physical perturbation...’, and so on.
Thanks for this comment. I should say here that the text of manuscript was edited my (Engish speaking) senior colleague Professor Barry Ninham, who is acknowleged as the world’ leading colloid and surface chemist, with over 500 research papers in many fields of сhemistry. However, in accordance with the reviewer's comments, the final version of the manuscript was re-edited by Barry. We hope that the latest version of the manuscript is free from the shortcomings associated with the presentation in English.
Line 41, ‘Teflon is very hydrophobic, while the sulfonate groups are very hydrophilic. On swelling in aqueous media, a nanostructure consisting of cylindrical reverse micelles forms. Water-filled channels 2 - 3 nm diameter form within the Nafion membrane’.
There is nothing wrong with the description here. However, as a cation exchange membrane material, ‘Nafion provides microchannels for cation conduction due to the phase separation of hydrophilic phase and hydrophobic phase under hydration conditions. The Teflon structure provides excellent mechanical and chemical stability, while the mobile protons produced by the dissociation of sulfonic acid groups provide active sites for cation conduction [Solid State Ionics 319 (2018): 110-116; Electrochimica Acta 378 (2021): 138133]’. These important details and cation conduction mechanisms should be briefly mentioned here.
We are grateful to the referee for this comment and the recommendation to refer to these articles. These articles are cited with appropriate comments in the latest version of the manuscript, see references [12] and [13].
Line 49, ‘This nanostructure is the key to spatial separation of Н+ and ОН– ions in low-temperature hydrogen power plants [3, 4].’
Is this due to the Donnan exclusion/repulsion effect? And in H2/O2 fuel cells, Nafion separates H+ and OH- from each half-cell as a separator. But finally, H+ may pass through the membrane and combine with OH- on the other side to form water. So the description should be modified to be more accurate. I do not consider H+ and OH- ions are separated forever and without any combination. Hence, the current description may lead to certain misunderstandings.
Thanks for this remark. In the new version we specially emphasize that H+ ion may pass through the membrane and combine with OH- ion on the other side to form water, so it is not correct to speak of a complete spatial separation of cations and anions on the membrane.
Line 96, ‘Those have a hydrophobic exterior.’ This short sentence should be merged with the former or the latter sentence.
This was done, thank you for this advice.
Line 101, ‘... we note that external influences on water and aqueous solutions of a pulsed electric field can lead to experimentally observed changes in the properties of water, see, e.g., [14].’ In the introduction, ‘see [reference number]’ appears multiple times. I suggest whether it is possible to briefly describe the general description found in other literature, otherwise the Introduction part provides too little effective information, and readers have to look through lots of literature to understand the research progress of related topics.
We have removed the reference to this article, because, indeed, it seems reasonable to give some comments on the results of the article cited. At the same time, the Introduction turns out to be too long, and mentioning the results of this article in the context of the subject of our work is not necessary.
Page 10 and Page 14, both the figures are called Figure 6, which is definitely the wrong name.
It was an annoying typo, thank you for this remark.
Line 318, ‘In Figure 7 we exhibit shows the results of measurements of the áCw(t)ñ value for...’Firstly, ‘exhibit shows’ only need one verb here. Secondly, Figure 7 demonstrates nothing about Cw(t), but Figure 6 on Page 14 shows the relevant results. I suggest the author check throughout the manuscript, whether the description of each figure in the main text is really matching the caption of the corresponding figure.
Thank you for this remark. All inaccuracies indicated by the reviewer have been eliminated in the latest version of the manuscript.
Line 356, ‘In order to obtain a numerical criterion for the difference between the studied salts and water.’ It seems this sentence is not completed yet, only half is described. In order to reach some targets, what is going to be done?
Thank you for this remark, the sentence was completed.
Reviewer 2 Report
Manuscript Draft ‘Nafion swelling in salt solutions in a finite sized cell: Curious phenomena dependent on sample preparation protocol’
In this manuscript authors have investigated the dependence of the properties of a salt solution (manly mixed aqueous solutions of NaCl, KCl, NaClO4 and NaClO3) on different behaviour of Nafion. Authors explored features of Nafion nanostructure in several electrolyte solutions that occur during the swelling is constrained. The strongly amphiphilic character of the polymer leads to a shear stress field and the expulsion of water from the complex swollen fiber mixture. During this process an air cavity is formed. The dynamics of the cavity collapse differs for solutions prepared by via different dilution protocols. Authors claims that these results are surprising. They may have implications for the standardization of pharmaceutical preparation processes.
The research idea is rather simple and just some basic investigations were performed. Nafion is rather often used for various technological purposes and swelling is rather well investigated Manuscript not significantly contributes to the field of polymer chemistry. Therefore, it is not recommended to publish the manuscript in MDPI Polymers.
Author Response
We thank referee for his comments. We, however, cannot fully agree with his assessment of our work. Indeed, in our work we have shown for the first time that the dynamics of Nafion swelling in an aqueous salt solution depends not only on the concentration of the solution (which is natural and quite expected), but also on the solution preparation protocol. This protocol includes a number of mechanical actions based on intense shaking and vortexing. We hypothesized that a certain role in changing the properties of a liquid is played by gas nanobubbles that appeared as a result of mechanical actions. Speaking about the effect of nanobubbles on the properties of aqueous electrolyte solutions, it is necessary to take into account the possibility of coalescence of nanobubbles, since the sizes of gas particles resulting from coalescence differ significantly from the sizes of nanobubbles. At the same time, for a number of electrolytes, there exists a critical concentration (0.17 M), above which coalescence is suppressed, while for other electrolytes, coalescence of nanobubbles occurs at any concentration. Thus, in our work we also studied the specific ionic effect in the context of changing the properties of the liquid. It is the study of the role of nanobubbles, taking into account the specific ionic effect that manifests itself under mechanical influences on a liquid sample and leads to a change in its properties, that our work is devoted to. In conclusion, we note that the influence of various mechanical actions on the properties of liquid solutions has recently been very intensively studied precisely in the context of the development of new technologies in pharmaceuticals.
Reviewer 3 Report
The manuscript suffers from several flaws:
- a too small outlining future applications of results found.
- There's no chemical scheme of Nafion.
- Pearson statistical coefficient should be computed for figure 4.
- Some graphs on figure 6 must be moved to the supplementary data file.
Author Response
We are grateful to the referee for a careful reading of the manuscript and very valuable comments. The manuscript was rewritten in accordance with the comments and recommendations of the reviewer. Below we present our responses to the reviewer's comments, which are in italics.
The manuscript suffers from several flaws:
- too small outlining future applications of results found.
Thanks for this comment. At the very end of the Conclusions section of the new version of the manuscript, we wrote: “Finally, we have shown that it is possible to change the physicochemical properties of salt solutions with the use of technologically processed solvent. This implies the theoretical possibility of controlling chemical reactions that occur in the body when using drugs. Thus, we can talk about new directions in pharmaceuticals”.
- There’s no chemical scheme of Nafion.
Thanks for this remark. This flaw was corrected. Figure 1 of the new version gives chemical structure of Nafion.
- Pearson statistical coefficient should be computed for figure 4.
We believe the reviewer had in mind the coefficients of variation for each experimental point in Fig. 4. These coefficients have been calculated. The results of these calculations have been added to the new version of the text.
- Some graphs on figure 6 must be moved to the supplementary data file.
We fully agree with the reviewer. The graphs for concentrations of 10^-2, 0.2 and 1 M are presented in manuscript-supplementary file; these are Fig_S1, Fig_S2 and Fig_S3 respectively.
Round 2
Reviewer 2 Report
Accept in present form